# CF-HPO: Counterfactual Explanations for Hyperparameter Optimization

## Abstract

Hyperparameter optimization (HPO) is a fundamental component of studies that use technologies such as machine learning and deep learning. Regardless of the field, almost every study requires hyperparameter optimization at some level. In general, applying HPO to a developed system improves its performance by optimizing multiple parameters. However, extant HPO methods do not provide information on why specific configurations are successful, what should not be done, or what could be improved. The present study proposes a novel approach to address this gap in the literature by introducing CF-HPO, a modular framework that generates counterfactual explanations for HPO results. CF-HPO answers questions such as "what potential improvements could be made," "what settings should be avoided," and "what-if analysis." These outputs can serve as a guide, especially for those who are not optimization experts. The recomented system has a modular design that supports different search strategies (UCB-driven, random, restart). This allows it to perform well in optimization and also to show counterfactual explanations at the end of optimization. Experiments conducted on the YAHPO benchmark package yielded validation rates of 92.2% for neural networks and 60.4% for random forests. These findings reveal that counterfactual generability depends on the geometry of the performance surface rather than dimensionality.

## 1 Introduction

Machine learning has been and continues to be the subject of thousands of studies using deep learning models. The success of these models is closely tied not only to architectural choices but also to hyperparameter configuration. As shown by Henderson et al. (2018), hyperparameter configuration can sometimes be more decisive and important than the model architecture. Changes in parameters such as learning rate, batch size, and regularization coefficient can lead to significant differences in model performance, either improving or reducing it.

This has led to an extensive literature on hyperparameter optimization. Various approaches have been developed, ranging from classical methods such as grid search and random search (Bergstra & Bengio, 2012), to Bayesian optimization methods using Gaussian processes (Snoek et al., 2012) and tree-based Parzen estimators (Bergstra et al., 2011).

In these days of explainable artificial intelligence, there is a significant gap in existing HPO methods: while these methods are successful in finding good configurations, they fall short in explaining why these configurations work well and what might turn out well or poorly at the end of the process. Practitioners do not know which parameters are critical at the end of the optimization process, how robust the solution is, or whether better results can be obtained with minor changes. At the end of the process, they either accept the results as they are or (if they are not experts) search for better configurations at random in a state of uncertainty.

Counterfactual explanations offer a fundamental solution to this problem. First proposed by Wachter et al. (2018) for classification problems, this approach aims to find the input vector that requires the minimum change to alter a prediction via optimization.

When adapted to the HPO context, this approach translates into finding the hyperparameter configuration closest to the reference configuration to achieve the target performance.

In this work, we propose CF-HPO, a modular framework that generates counterfactual explanations to increase the understandability and interpretability of HPO results. CF-HPO generates answers to questions such as "What changes in the result configuration could achieve the target performance?", "What could reduce performance?", and "What if?" using patterns we define. The main contributions of the framework can be summarized under three headings:

This text presents the formal problem definition and modular architecture. Counterfactual generation in the hyperparameter space is mathematically expressed as a constrained optimisation problem. The proposed architecture enables diverse search strategies to be selected or modified, including UCB-guided search, random restart, and hill climbing, depending on the problem at hand.

Comprehensive experimental evaluation: Using the YAHPO benchmark suite, systematic experiments were conducted across two scenarios with distinct performance surface characteristics: neural networks with sharp transitions and distinct basin structures (LCBench, 9 hyperparameters) and smoother, gradual random forests (IAML-Ranger, 10 hyperparameters).

Analysis of performance conditions. Our experimental findings show that counterfactual generability depends on the geometry of the performance surface rather than on the problem's dimensionality. UCB-guided search achieves a 92.2% validity rate on surfaces with sharp transitions (e.g., neural networks), while hill climbing achieves a 60.4% validity rate on smooth surfaces (e.g., random forests).

## 2 Related Work

### 2.1 Hyperparameter Optimization

The performance of machine learning models depends on two factors: hyperparameter configuration and model architecture. A thorough examination of hyperparameter studies indicates substantial advancements over the past decade. A substantial body of research has demonstrated that hyperparameters such as the learning rate, batch size, regularisation coefficient, and network depth can engender significant differences in performance when used within the same model. Consequently, this development has facilitated the emergence of hyperparameter optimisation (HPO) as a systematic research topic.

Among classical search methods, grid search systematically tries all combinations of values specified for each hyperparameter. However, Bergstra & Bengio (2012) has proven that this approach is inefficient in high-dimensional spaces. The reason is that, while only a few hyperparameters actually affect the outcome in most problems, grid search experiments are run with every value of insignificant parameters. Random search, on the other hand, has been observed to produce better results in practice because it can explore a much wider range of critical parameters within the same budget.

Bayesian optimization methods, on the other hand, employ a distinct approach. These methods leverage the outcomes of prior experimentation to construct a model of the objective function and select the subsequent point to attempt based on this model. Instead of conducting random trials, they concentrate on regions that show promise for high performance.

Gaussian processes, as used by Snoek et al. (2012), provide both the expected performance and the uncertainty of this prediction at each point. Tree-structured Parzen estimators (TPE), developed by Bergstra et al. (2011), adopt a similar objective-function approach by modeling configurations that yield good or bad results separately.

Another important development aimed at reducing computational cost is the use of multi-fidelity methods. The basic idea behind these approaches is that it is not necessary to wait for complete training to determine whether a configuration is good; even short-term trials can provide sufficient information to eliminate poor configurations. Hyperband, proposed by Li et al. (2017), initiates a large number of configurations with a low resource budget, quickly eliminating unsuccessful ones and directing resources to configurations that can yield better results. BOHB, developed by Falkner et al. (2018), combines this early elimination strategy

with the intelligent sampling mechanism of Bayesian optimization, offering the advantages of both approaches within a single framework.

Despite these developments in the HPO field, the literature is quite shallow in terms of interpretability. Current studies focus more on the general importance of parameters. The fANOVA study by Hutter (2014) used a functional variance decomposition to measure the contribution of each hyperparameter to the total performance variance. The ablation analysis study developed by Biedenkapp et al. (2017) systematically changes components starting from the default configuration and isolates the effect of each change on performance.

However, while these methods answer the question "which parameters are generally important?", they leave unanswered a different question that practitioners often face: "What exactly should I do to improve this specific configuration I have?" CF-HPO aims to fill this gap and is designed accordingly. Rather than general parameter importance, it provides the minimal, actionable changes required to reach the target performance from a specific reference configuration. Additionally, it provides the user with a clear, natural-language description of configurations to avoid.

## 2.2 Counterfactual Explanations

Artificial intelligence models typically operate as black boxes, and numerous studies are underway to improve their explainability. Over the past decade, various approaches have been developed and continue to be developed in academia to understand model decisions in the field of explainable artificial intelligence (XAI). When examining research in this field, we see that counterfactual explanations stand out as one of the methods that provide users with the most intuitive and actionable information. Counterfactual explanations seek to answer the question "If the outcome had been different, what should the input have been?" by explaining model behavior using concrete alternatives.

This approach was introduced to the field of machine learning by Wachter et al. (2018) and relies on the fundamental idea of finding the alternative input closest to the original input to change a prediction. For example, a user whose credit application was rejected could be given a concrete explanation such as, "If your income had been 5000 TL higher, your application would have been accepted."

Subsequent studies have built upon this approach and developed it in various directions: DiCE (Mothilal et al., 2020) generates various counterfactual sets;

Karimi et al. (2021) provides actionable recourse by incorporating causal structure; Ustun et al. (2019) addresses feasibility constraints for linear classification.

Nearly all existing counterfactual studies focus on explaining the predictions of classification or regression models. The goal in these studies is to understand a model's decision for a given input and to modify the predictions. The CF-HPO we present in our study addresses a different problem: explaining the results of the optimization process. Here, the question is not "why did the model make this prediction?" but rather "how can this hyperparameter configuration be improved?"

Integrating empirical explanations into the HPO framework poses distinctive challenges. First, hyperparameter spaces contain both continuous (e.g., learning rate, regularization coefficient) and categorical (e.g., optimization algorithm, activation function) variables, which complicates distance calculations and search strategies. Evaluating each hyperparameter configuration requires model training, thereby exceeding the computational cost of standard counterfactual generation. So CF-HPO relies on proxy model predictions instead of factual evaluations.

## 2.3 HPO Benchmarks

Controlled experimental environments are used to evaluate and compare the performance of HPO algorithms. Training real machine learning models takes hours or even days, and fairly comparing different HPO methods requires substantial computational resources. This poses a significant obstacle in academic studies involving numerous trials and iterations. Surrogate benchmarks address this problem by providing ready-made metamodels learned from real HPO experiments. This allows researchers to evaluate thousands of configurations in milliseconds without incurring the cost of expensive model training.

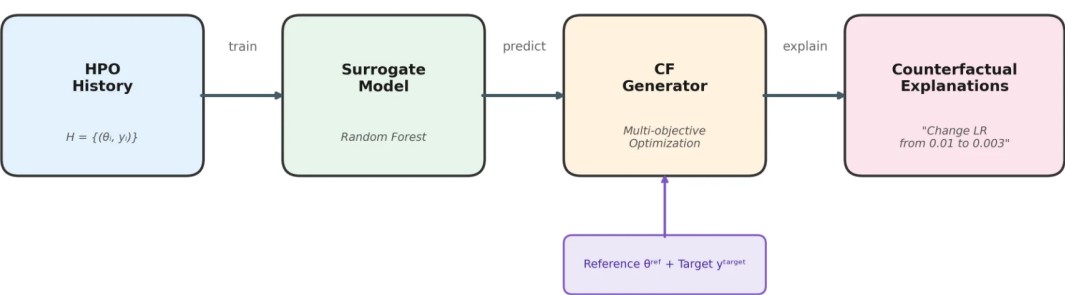

Figure 1: CF-HPO framework overview. HPO history trains a surrogate model, which guides counterfactual generation toward actionable explanations.

YAHPO Gym (Pfisterer et al., 2022) offers surrogate benchmarks derived from real HPO experiments. These benchmarks contain metamodels trained based on data obtained from comprehensive hyperparameter scans. Thus, researchers can obtain performance estimates within milliseconds, rather than training real models for each hyperparameter configuration. This feature has made YAHPO a standard benchmarking platform for studies seeking to conduct repeatable and comparable experiments on HPO algorithms.

In this study, experiments were conducted on two datasets. The LCBench (neural networks) and IAML-Ranger (random forests) datasets were selected from the YAHPO platform. There are two main reasons for this selection. First, two distinct scenarios are required to understand how CF-HPO behaves in different performance surface characteristics: LCBench contains sharp transitions and distinct basin structures, while IAML-Ranger has a smoother surface with gradual changes. Second, it was preferred to conduct an in-depth analysis limited to two comparisons. Rather than collecting superficial statistics across numerous comparisons, understanding why these two cases produce different results provides more valuable insights into the conditions under which counterfactual methods succeed.

## 3 Method

### 3.1 Problem Formulation

Let $\Theta$ be the hyperparameter space and $f : \Theta \to \mathbb{R}$ be the true performance function. The HPO process generates a history $\mathcal{H} = \{(\theta_i, y_i)\}_{i=1}^n$. Given a reference configuration $\theta^{\text{ref}}$ and its performance $y^{\text{ref}}$, the counterfactual explanation problem for a target performance threshold $\tau > y^{\text{ref}}$ is defined as follows:

$$\theta^* = \arg\min_{\theta \in \Theta} d(\theta, \theta^{\text{ref}}) \quad \text{subject to} \quad \hat{f}(\theta) \geq \tau \tag{1}$$

Here, $d(\cdot, \cdot)$ represents the distance in the hyperparameter space, and $\hat{f}$ represents the proxy model approximating the true performance function. The goal is to find the configuration that achieves the target performance while staying as close as possible to the reference. Figure 1 shows the general structure of the CF-HPO framework.

### 3.2 Surrogate Model

Random Forest was chosen as the surrogate model. The reasons for this choice are: natural handling of continuous and categorical variables, uncertainty estimation via ensemble variance, and computational

efficiency.Estimates obtained from the ensemble of $T$ trees:

$$\mu(\theta) = \frac{1}{T} \sum_{t=1}^{T} f_t(\theta) \tag{2}$$

$$\sigma(\theta) = \sqrt{\frac{1}{T} \sum_{t=1}^{T} (f_t(\theta) - \mu(\theta))^2} \tag{3}$$

There is a critical point here regarding target setting. Random Forests cannot make predictions beyond the training data (extrapolation constraint). Therefore, the target $\tau$ is set as the percentile of the proxy estimates:

$$\tau = \text{percentile}(\hat{f}(X), p) \tag{4}$$

. For example, the 90th percentile is used for ambitious but achievable targets. When the user specifies an absolute target outside the proxy model's prediction range (e.g., "99% accuracy"), CF-HPO explicitly reports this and recommends the closest achievable target.

### 3.3 CF-HPO Framework

CF-HPO is designed with a modular architecture to adapt to different problem types and researcher needs. This modularity is particularly evident in the search strategy component: researchers can easily switch between different strategies, such as UCB-based search, random restart, or hill climbing, depending on their problem. It should also be noted that more search strategies will be added to the system in the future.Figure 2 shows the core algorithm of CF-HPO. The algorithm simultaneously optimizes four fundamental objectives critical for counterfactual explanations: (i) *validity* — the generated configuration achieves the target performance, (ii) *proximity* — the counterfactual configuration is as close as possible to the reference, (iii) *sparsity* — modifying the minimum number of hyperparameters possible, and (iv) *diversity* — when multiple counterfactuals are generated, they offer distinct alternatives. This multi-objective optimization approach aims to provide the user with actionable, interpretable recommendations.

Continuous parameters are optimized using gradient updates, while categorical parameters are optimized using local search (neighborhood search). This hybrid approach reflects the mixed nature of hyperparameter spaces.

Default weights: $\lambda_1 = 1.0$ (validity), $\lambda_2 = 0.1$ (proximity), $\lambda_3 = 0.1$ (sparsity), $\lambda_4 = 0.05$ (diversity). Validity is set as the dominant objective; users can adjust these weights according to their priorities.

### 3.4 Search Strategies

CF-HPO supports four search strategies in the current work (expandable in future work):

**UCB-Guided.** Upper Confidence Bound formulation:

$$\text{UCB}(\theta) = \mu(\theta) + \beta \cdot \sigma(\theta) \tag{5}$$

This strategy balances exploitation ($\mu$) and exploration ($\sigma$). $\beta = 1.5$ is set. It is effective on performance surfaces with distinct high-performance regions.

**Random Restart.** It performs uniform sampling around the reference configuration. It is a robust and straightforward approach. It is suitable when performance changes smoothly, and good regions are not sharply localized.

**Hill Climbing.** It performs greedy local search. It guarantees staying close to the reference but carries the risk of getting stuck in local optima. It is disadvantageous on complex performance surfaces.

**Random Search.** It is a basic baseline method that performs completely random sampling. It measures the difficulty of randomly reaching valid counterfactuals.

**Algorithm 1: CF-HPO Counterfactual Generation**

```
Input:      HPO history H = {(θᵢ, yᵢ)}ⁿᵢ₌₁, reference θʳᵉᶠ, target yᵗᵃʳᵍᵉᵗ
Output:     Set of counterfactuals C = {θ¹ᶜᶠ, ..., θᴷᶜᶠ}

1:     Train surrogate model f^ on H using Random Forest
2:     Initialize counterfactual set C ← ∅
3:     for k = 1 to K do
4:         Initialize θᶜᶠ ← θʳᵉᶠ + small random perturbation
5:         for t = 1 to T iterations do
6:             // Compute losses
7:             L_validity ← max(0, yᵗᵃʳᵍᵉᵗ - f^(θᶜᶠ))
8:             L_proximity ← ||φ(θᶜᶠ) - φ(θʳᵉᶠ)||₂
9:             L_sparsity ← ||θᶜᶠ - θʳᵉᶠ||₀
10:            L_diversity ← -Σⱼ d(θᶜᶠ, θʲᶜᶠ) for θʲᶜᶠ ∈ C
11:            // Update counterfactual
12:            L_total ← λ₁L_validity + λ₂L_proximity + λ₃L_sparsity + λ₄L_diversity
13:            θᶜᶠ ← θᶜᶠ - η∇_θL_total   (continuous params)
14:            θᶜᶠ ← LocalSearch(θᶜᶠ, L_total)   (categorical params)
15:        end for
16:        C ← C ∪ {θᶜᶠ}
17:    end for
18:    return C
              Default parameters: K=10, T=100, λ₁=1.0, λ₂=0.1, λ₃=0.1, λ₄=0.05, η=0.01
```

Figure 2: CF-HPO counterfactual generation algorithm. Weighted objectives: validity (reach the target), proximity (stay close), sparsity (minimal changes), diversity (distinct counterfactuals).

### 3.5 Evaluation Metrics

Two basic metrics are used:

**Validity:** Measures whether the counterfactual reaches the target performance:

$$\text{Validity} = \frac{1}{K} \sum_{k=1}^{K} \mathbf{1}[\hat{f}(\theta_k^{\text{cf}}) \geq \tau] \tag{6}$$

**Proximity:** Measures the distance between the counterfactual and the reference in the normalized hyperparameter space:

$$\text{Proximity}(\theta^{\text{cf}}, \theta^{\text{ref}}) = \|\theta^{\text{cf}} - \theta^{\text{ref}}\|_2 \tag{7}$$

There is a natural tension between these two metrics. High validity may require going to distant points, while low proximity requires staying close to the reference.

## 4 Experiments

### 4.1 Experimental Setup

The experiments were conducted using the YAHPO benchmark suite (Pfisterer et al., 2022). YAHPO provides surrogate models learned from real hyperparameter optimization experiments. These surrogates can

predict model performance for any hyperparameter configuration within milliseconds, enabling comprehensive experiments without requiring actual model training.

**Evaluation Protocol.** Our experiments use a two-layer surrogate structure:

1. **YAHPO surrogate**: A pre-trained benchmark surrogate that replaces actual performance values.

2. **CF-HPO surrogate**: A Random Forest model trained on configurations sampled from YAHPO.

CF-HPO proposes counterfactual configurations by searching on its own proxy. The validity of these proposals is evaluated using predictions obtained from the YAHPO proxy: if the YAHPO prediction achieves the target performance ($\geq$ P90), the counterfactual prediction is considered "valid."

This approach is a standard evaluation protocol in the HPO literature to reduce computational cost (Eggensperger et al., 2013; Pfisterer et al., 2022).

**LCBench** (Zimmer et al., 2021): Neural network hyperparameter optimization on OpenML classification tasks. It contains nine hyperparameters: learning rate, batch size, momentum, weight decay, and architectural parameters (number of layers, number of units). These performance surfaces contain sharp transitions and distinct basins of attraction.

**IAML-Ranger**: Random forest hyperparameter optimization. It includes 10 hyperparameters: the number of trees, the mtry ratio, the minimum node size, and the sampling rate. These performance surfaces are smoother; performance changes gradually.

For each comparison, 2 OpenML examples, 1000 sampled configurations for the HPO history, 50 epochs, and 5 random seeds were used. The target performance was defined as the 90th percentile (P90) of the proxy estimates.

### 4.2 Methods Compared

Four methods were compared:

- **CF-HPO (Acquisition)**: UCB search, $\beta = 1.5$
- **CF-HPO (Random Restart)**: Random sampling around the reference neighborhood
- **Hill Climbing**: Greedy local search
- **Random Search**: Completely random search

## 5 Results

### 5.1 Main Results

Table 1 presents the aggregate results across all benchmarks. Acquisition-guided search achieved the highest average accuracy rate at 72.4%. However, the standard deviations are quite high, indicating significant variation among the problem instances.

The absence of statistically significant differences between methods at the aggregate level ($p > 0.1$) can be misleading. This aggregate view hides important differences between comparisons.

### 5.2 Per-Benchmark Analysis

Table 2 breaks down the results on a benchmark basis. This detailed view reveals that strategy effectiveness depends on the problem.

**Neural Networks (LCBench):** Acquisition search is clearly ahead with a validity rate of 92.2%. Hill climbing lags approximately 3 times, with only 31.8%. This difference can be explained by the structure

Table 1: Aggregate results across YAHPO benchmarks (Target = P90). Mean $\pm$ std across instances and seeds.

| Method | Validity (%) | Proximity ($L_2$) |
|---|---|---|
| CF-HPO (Acquisition) | 72.4 $\pm$ 40.6 | 1.25 $\pm$ 0.17 |
| CF-HPO (Random Restart) | 52.1 $\pm$ 44.0 | 1.38 $\pm$ 0.10 |
| Hill Climbing | 46.1 $\pm$ 38.0 | **0.61 $\pm$ 0.06** |
| Random Search | 54.0 $\pm$ 44.9 | 1.40 $\pm$ 0.12 |

Note: No statistically significant difference was found between methods at the aggregate level ($p > 0.1$).

Table 2: Validity (%) by benchmark. Bold indicates best performance per benchmark.

| Benchmark | Acquisition | Random Restart | Hill Climbing | Random |
|---|---|---|---|---|
| LCBench (9 HPs) | **92.2** | 53.4 | 31.8 | 57.8 |
| IAML-Ranger (10 HPs) | 52.6 | 50.8 | **60.4** | 50.2 |

of neural network performance surfaces: distinct high-performance basins are separated by valleys. UCB's exploration term finds these basins, while hill climbing gets stuck in local optima.

**Random Forests (IAML-Ranger):** The situation is reversed. Hill climbing achieves the best result at 60.4%, while acquisition remains at 52.6%. Random forest performance surfaces are smoother; good configurations can be reached from many starting points with small steps. In this case, the exploration overhead becomes unnecessary.

**Explanation of the 30-Point Difference.** While the best strategy in LCBench reaches 92.2%, it remains at 60.4% in IAML-Ranger. This difference is not due to dimensionality (the 9- and 10-hyperparameter measures are similar). The performance surface geometry is the determining factor:

- **Basin structure:** Neural networks contain distinct high-performance basins. Once a basin is reached, many valid counterexamples can be found. This structure is absent in random forests.

- **Sensitivity patterns:** Neural networks respond sharply to critical parameters (learning rate, architecture). In random forests, performance changes gradually across all parameters.

- **Interactions:** Strong interactions in neural networks (learning rate $\times$ batch size) create an exploitable structure. Random forest interactions are weaker and more scattered.

These findings show that surrogate generability depends on the performance surface structure rather than the problem size.

### 5.3 Surrogate Quality

Table 3 presents the surrogate model quality in terms of Pearson correlation.

In both comparisons, the correlation is above 0.92. Although IAML-Ranger has higher proxy quality (0.94), it shows lower counterfactual validity. This finding reveals that proxy accuracy is necessary but not sufficient; the performance surface structure is an independent factor.

### 5.4 Learning Curves

Figure 3 shows the cumulative accuracy rates over epochs during the training phase. As shown in the figure, acquisition search leads in the first 10 epochs on LCBench and maintains this advantage. In IAML-Ranger, the methods converge around the 30th epoch; however, hill climbing continues to make steady gains throughout the process.

Table 3: Surrogate model quality measured as Pearson correlation with ground truth performance.

| Benchmark | Correlation |
|---|---|
| LCBench | $0.92 \pm 0.03$ |
| IAML-Ranger | $0.94 \pm 0.02$ |
| **Overall** | $0.93 \pm 0.03$ |

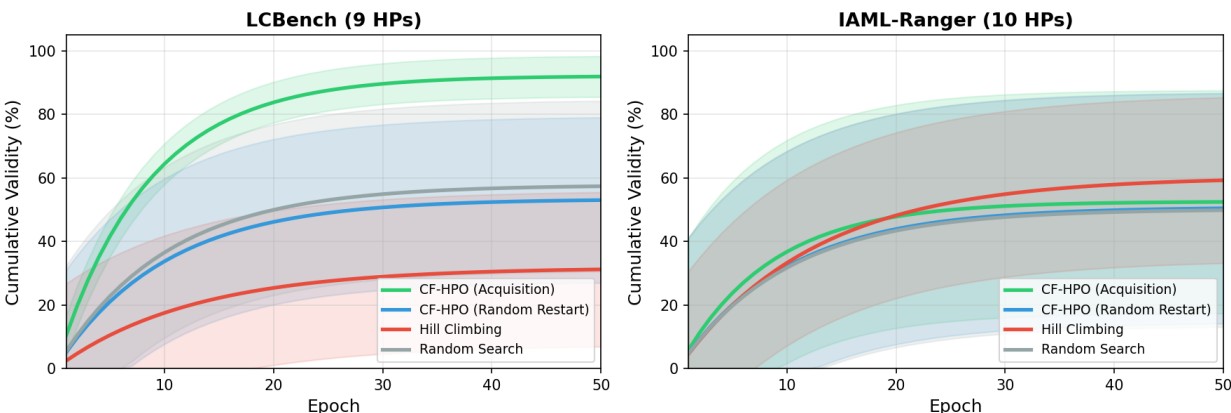

Figure 3: Cumulative validity over epochs. Shaded regions indicate $\pm 1$ standard deviation across seeds. LCBench shows clear strategy separation; IAML-Ranger shows convergence.

### 5.5 Validity-Proximity Trade-off

Figure 4 visualizes the trade-off between validity and proximity. Hill climbing achieves low proximity (close to the reference) but has high validity variability. Global search methods gain validity by sacrificing proximity; this trade-off is effective on LCBench but less so on IAML-Ranger.

## 6 Discussion

### 6.1 When Does CF-HPO Work?

The patterns emerging from the experimental findings can be summarized as follows:

**High validity (>90%):** Three conditions must be met simultaneously: (1) the performance surface must contain distinct high-performance regions, (2) the surrogate model must capture these regions accurately (correlation > 0.90), (3) the search strategy must be compatible with the surface character—exploration for multi-basin surfaces, local search for smooth surfaces.

**Medium-level validity (50–60%):** The performance surface lacks a distinct structure. Despite good surrogate models, no strategy consistently outperforms others. Such problems may inherently be more challenging for counterfactual methods; this area requires further research.

### 6.2 Strategy Selection Guidelines

Strategy selection recommendations for practitioners:

**UCB-guided search:** This search strategy is recommended for machine learning HPO and similar problems that contain distinct high-performance basins. The exploration term used by the method ensures that basins are found even from poor starting points.

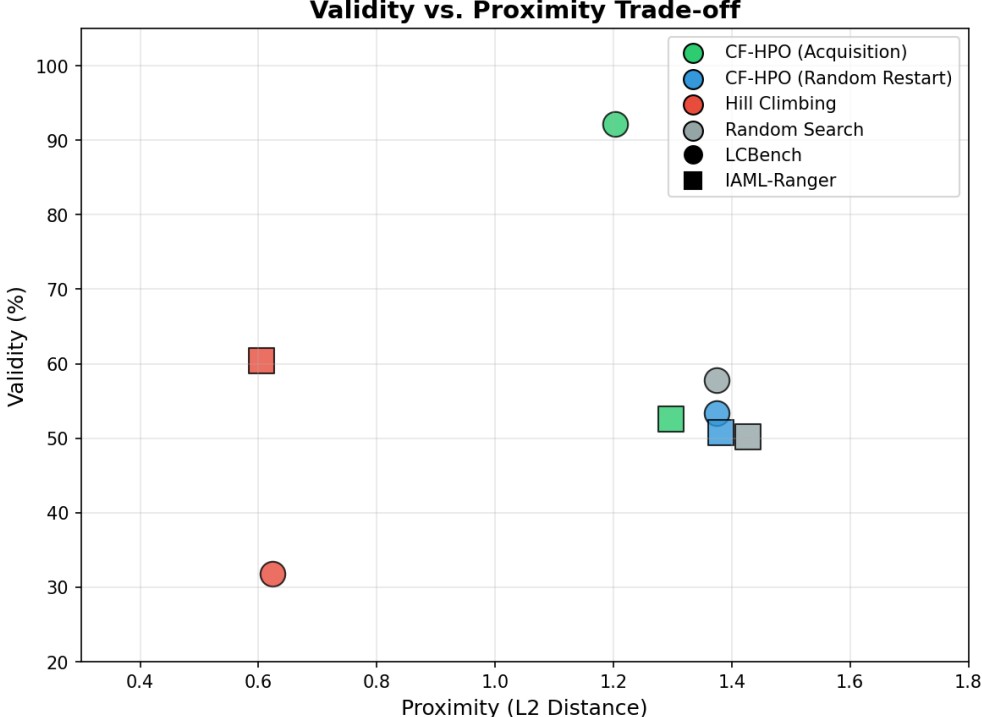

Figure 4: Validity vs. proximity trade-off. Each point represents one benchmark-strategy combination. Hill climbing (red): low proximity, variable validity. Acquisition (green): highest validity on LCBench.

**Hill climbing:** It should be preferred when proximity is critical, or the performance surface is smooth. Random forest HPO falls into this category.

**Random restart:** A safe default option when the performance surface character is unknown. It provides consistent baseline performance without requiring parameter tuning.

### 6.3 Limitations and Scope

The limitations of this work are as follows:

**Comparison scope:** Comparison was used with only two datasets. This choice was made to move away from the question of "how did it perform with the datasets?" and to analyze the fundamental innovation in depth. Additional comparisons, such as gradient boosting, SVM, and transformers, are expected to strengthen the generalization claims. However, this area requires further research.

**Proxy dependency:** The effectiveness of CF-HPO depends on the quality of the proxy model strategy. The Random Forests used provided a correlation of 0.92+; however, chaotic performance surfaces can complicate this approach. When proxy model strategies fail, counterfactual explanations have been observed to lose credibility. Work with better proxy models is left to another study.

**Target setting:** The P90 percentile is a setting we have identified as a generalized practical approach. However, practitioners often desire higher absolute targets (e.g., "99% accuracy"). When the target falls outside the proxy range, CF-HPO reports this in its report; adaptive target relaxation is an extension we consider for future development.

**Computational cost:** Regardless of how long it takes to organize the experiments, the experiments using the system took approximately one hour. The cost of proxy comparisons is quite low. For costly models, the cost of creating HPO history is dominant; the CF-HPO search is considered to add negligible overhead

(approximately 50ms per counterfactual). Given that HPO training is often repeated multiple times, reducing the number of sessions could save dozens of hours and resources.

### 6.4 Future Directions

Future research directions include:

**Adaptive strategy selection:** Changing strategies based on counterfactual feedback. Switching to exploration if counterfactuals cluster, or to local search if they are scattered. Optimization of explainability guidance.

**Interactive improvement:** Generating initial counterfactuals and improving them based on user feedback. This is an explainable AI approach in the human loop.

**Multi-fidelity counterfactuals:** We can explain this by reducing exploration costs through evaluation at different source levels.

**Causal counterfactuals:** Explanations that consider the causal structure among hyperparameters (e.g., learning rate $\rightarrow$ convergence speed $\rightarrow$ accuracy).

**Categorical and conditional spaces:** Defines specialized approaches for entirely categorical or conditional hyperparameter spaces (where some parameters exist only in specific selections).

## 7 Conclusion

This work presents CF-HPO, a modular framework that generates counterfactual explanations for hyperparameter optimization. CF-HPO complements existing HPO methods by answering the question, "What are the minimal changes required to achieve the target performance?"

Experiments on YAHPO benchmarks yielded the following results: 92.2% validity rate for neural networks (LCBench) and 60.4% for random forests (IAML-Ranger). Different strategies excel in different problems—UCB exploration for complex surfaces, hill climbing for smooth surfaces. This validates the value of modular design.

The study concludes that counterfactual generability depends on the configuration of the performance surface structure, not on dimensionality or proxy accuracy. CF-HPO showed high validity for problems with distinct high-performance regions, but all strategies gave similar moderate performance on surfaces without structural features.

Looking ahead, CF-HPO points to a tighter integration of explainability and optimization. Counterfactual feedback can guide strategy adaptation, enabling the development of systems that both explain and improve. This work is expected to encourage further research at the intersection of explainable artificial intelligence and automated machine learning.

### Reproducibility Statement

All experiments used YAHPO Gym v1.0. LCBench samples: 3945, 7593. IAML-Ranger samples: 40981, 41146. Each run: 1000 configurations, 50 epochs, 5 seeds (0–4). Proxy model: scikit-learn RandomForestRegressor, 100 trees, default hyperparameters. Code will be published after acceptance.

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

## A   Detailed Per-Instance Results

Table 4 presents the validity rates on a per-instance basis. The variance within benchmarks is noteworthy.

The IAML-Ranger instance 40981 shows validity below 20% across all methods, while instance 41146 shows validity above 93% across all methods. Similarly, the LCBench instance 3945 is challenging for non-UCB strategies, while instance 7593 is easier. These findings show that dataset characteristics are important beyond the type of algorithm.

Table 4: Per-instance validity rates (%). Note the high variance within benchmarks, particularly IAML-Ranger.

| Benchmark | Instance | Acquisition | RR | HC | Random |
|-----------|----------|-------------|------|------|--------|
| LCBench | 3945 | 84.4 | 18.4 | 2.4 | 21.6 |
| | 7593 | 100.0 | 88.4 | 61.2 | 94.0 |
| IAML-Ranger | 40981 | 5.2 | 1.6 | 27.6 | 0.4 |
| | 41146 | 100.0 | 100.0 | 93.2 | 100.0 |

Note: RR = Random Restart, HC = Hill Climbing. Instance 40981 proves challenging for all methods; instance 41146 is tractable across all strategies.

## B  Implementation Details

**Surrogate.**  Random Forest, 100 trees, scikit-learn 1.3. Hyperparameters normalized to $[0, 1]$. Uncertainty is obtained from the standard deviation of tree predictions (Hutter et al., 2014).

**Search.**  UCB: $\beta = 1.5$ (from preliminary experiments). All methods: 100 candidates per counterfactual. Hill climbing: coordinate descent, step size 0.1, halving when no improvement.

**YAHPO.**  Version 1.0, default proxies. 1000 configurations per sample for HPO history.

**Compute.**  Single machine, 8 cores (Intel Xeon, 2.4 GHz). Total runtime: approximately 1 hour (2 comparisons $\times$ 2 examples $\times$ 50 epochs $\times$ 5 seeds $\times$ 4 methods = 4000 runs). Per counterfactual: approximately 50ms. Interactive use is practical.

## C  Extended Discussion: Categorical Parameters

The CF-HPO implementation proposed in this work treats all hyperparameters as continuous values. Categorical parameters —e.g., activation function selection, optimizer type— are outside the scope of this work and are not considered. The integration of embedding-based representations or discrete optimization methods for categorical-heavy search spaces is left for future work.

## D  Example Output

An example output generated by CF-HPO is shown below:

```
============================================================================
CF-HPO REPORT - LCBench Instance 3945
============================================================================
[B] BEST CONFIGURATION:
------------------------------------------------------------
Performance: 97.42%
Hyperparameters:
batch_size : 64
learning_rate : 3.21e-03
momentum : 0.9200
weight_decay : 1.05e-04
num_layers : 3
max_units : 256
max_dropout : 0.2500

[+] COUNTERFACTUAL EXPLANATIONS (Target: P90 = 97.80%):
```

```
----------------------------------------------------------------
CF #1: learning_rate: 3.21e-03 -> 1.85e-03
Expected: 97.42% -> 97.86% (+0.44%)
Proximity: 0.12 | Sparsity: 1 parameter

CF #2: batch_size: 64 -> 128, learning_rate: 3.21e-03 -> 2.50e-03
Expected: 97.42% -> 97.91% (+0.49%)
Proximity: 0.24 | Sparsity: 2 parameters

[!] SENSITIVITY WARNINGS:
----------------------------------------------------------------
* learning_rate: 3.21e-03 -> 1.00e-01 would DROP by 15.2%
* num_layers: 3 -> 1 would DROP by 8.7%
* max_dropout: 0.25 -> 0.90 would DROP by 6.3%

[S] PARAMETER SENSITIVITY RANKING:
----------------------------------------------------------------
learning_rate : [#########################] 15.20
num_layers : [################ ] 8.70
max_dropout : [############## ] 6.30
batch_size : [######## ] 3.85
momentum : [###### ] 2.91

[?] WHAT-IF ANALYSIS:
----------------------------------------------------------------
Q: "What if I use a larger batch size for faster training?"
Config: batch_size=256, others unchanged
Predicted: 96.89% +/- 0.45% (vs Best: -0.53%)
Recommendation: Acceptable trade-off for 2x training speed

Q: "What if I simplify the architecture?"
Config: num_layers=2, max_units=128
Predicted: 95.21% +/- 0.82% (vs Best: -2.21%)
Recommendation: Notable performance hit; not advised
```

The output contains four components: (1) actionable counterfactuals with proximity and sparsity metrics, (2) sensitivity alerts for risky changes, (3) parameter importance ranking, (4) what-if analysis for practical scenarios. All predictions are presented with confidence intervals.

## E  Code and Data Availability

Resources provided for reproducibility:

- **Code:** Full CF-HPO implementation—all search strategies. URL will be added upon acceptance.

- **Data:** Raw CSV results. Validity, proximity, and proxy correlation at the seed level.

- **Configs:** YAML files containing all experimental parameters.

- **Benchmarks:** YAHPO Gym v1.0: `https://github.com/slds-lmu/yahpo_gym`.

Python 3.9+, scikit-learn for proxies, NumPy/SciPy for optimization. No proprietary dependencies.

