# OpenReview forum: "CF-HPO: Counterfactual Explanations for Hyperparameter Optimization"
_TMLR — Rejected by TMLR_

### Review · Reviewer_FonQ · 2026-03-13

**Summary Of Contributions:**

The paper "CF-HPO: Counterfactual Explanations for Hyperparameter Optimization" basically does what the title states: apply counterfactual explanations to hyperparameter optimization outcomes. The paper proposes to use gradient-based multi-objective search for counterfactuals (and a nearest neighbor-based search for discrete hyperparameters), where the different objectives are weighted by manually set weights.

## Strengths:
* Novel idea: to the best of my knowledge, no one has used counterfactuals to explain hyperparameters yet.
* Significance: if this idea works, it will have great impact on hyperparameter optimization interpretation, because, as argued, it could provide a more intuitive way to interpret hyperparameters.

## Weaknesses:
The paper suffers from substantialy weaknesses in the methodology, the experimental setup and the writeup. Also, the results are rather weak, which does not support the idea of using this explanation method in practice.

**Audience:**

Yes

**Audience Explanation:**

The main strength of this work is the possibility to better explain hyperparameters in a language a human would understand: if you make this change, your performance will improve. Hyperparameter optimization and interpretation of hyperparameters are important topics that will be of interest to parts of TMLR's readership.

**Claims And Evidence:**

No

**Claims Explanation:**

Unfortunately, I see a major issues with the methodology and implementation of the proposed CF-HPO. Second, I think the experiments are executed in a suboptimal fashion. Finally this, the quality of the write-up is not too high, which I will also explain.

## Issues in the methodology and implementation

1. Equation 4 is not defined thoroughly. I do not understand with respect to which values the percentile is calculated. Are these the observations made during the hyperparameter optimization process? Following up on this, what is the implication of using a 90th percentile? The paper states this is for "ambitious, but achievable targets", but does not further discuss why achieving something that is 90% as good as the best result is desirable. I think an evaluation of the empirical CDF of the observed function evaluations would be warranted in order to explain this to the reader.
2. If I interpret Equation 4 as being dependent on the observed hyperparameter values, this explains why the proposed method does not work well for random search. The area around the best configuration is most likely not explored very well, leading to high noise.
3. The paper uses a weighting-based multi-objective formulation of the counterfactuals generation without giving a reason why this would be a good setting. A more general setting that does not require the objectives to be scalarized is presented in "Multi-objective counterfactual fairness" by Dandl, Pfisterer and Bischl (GECCO 2022 https://dl.acm.org/doi/10.1145/3520304.3528779)
4. The pseudo-code does not match the description of the algorithm. It does not contain the percentiles introduce in Equation 4, and contains an issue in Line 7: there is no gradient with respect to the validity if we the prediction according to the surrogate is worse than the reference solution. Therefore, in practice, since the used random forest will not extrapolate (as actually explained), the algorithm will not optimize for validity.

## Issues in the experimental setup

I appreciate that the papers aims to focus on only two benchmarks problems, and then analyze them thoroughly. Unfortunately though, the thorough analysis does not happen. The paper claims that the landscapes of the optimization problems has an influence on the results, but no analysis of the landscape is made in the paper nor cited.

Another big issue with this paper is that the method very often only achieves a validity of <80%. This means, that more than 20% of the explanations are wrong. I am wondering if this is really an explanation method that should be proposed at all. Would it not make sense to rather figure out the area around the best value that are within a certain performance (Rashomon set) to explain the effect of the hyperparameters?

Minor issues:
* What test is performed in Section 5.2?

## Issues in the writeup

* Sometimes, the paper writes "parameter" instead of "hyperparameter".
* Related work is referenced very sparingly, for example, a reference to YAHPO and LCBench should be added where they are cited initially in the introduction.
* Sometimes, claims are not backed by references at all:
    1. First paragraph of Section 2.1
    2. First few sentences of Section 2.2
* Sometimes, relevant concepts are explained, but not cited properly, for example:
    * Bayesian optimization: I would have assumed a reference to the book from Roman Garnett.
    * Surrogate benchmarks for hyperparameter optimization: I would have assumed a reference to the papers from Eggensperger et al. at AAAI 2015 and the Machine Learning Journal.
* Bergstra and Bengio did not prove that random search is more efficient than grid search (as claimed in the 2nd paragraph of Section 2.1), but rather demonstrated this and gave convincing explanations for the fact.
* The explanation of Bayesian optimization in Section 2.1 is rather short and superficial: it does not discuss main features of Bayesian optimization, such as the uncertainty quantification in the underlying model and the exploration/exploitation tradeoff.
* There is a linebreak on Page 3 after a semicolon, where there should be none.
* Several sentence ends are not followed by a whitespace.
* Certain algorithms are not explained, for example:
    * "local search (neighborhood search)" in Section 3.3
    * UCB in Section 3.4
    * Random Restart in Section 3.4. This misses a definition of "uniform sampling around the reference configuration", i.e., how far around it samples.
    * Similarly, there is no description of how hill climbing is implemented
* The placement of the default parameters in the pseudocode is suboptimal. Also, the parameter T is not properly defined.
* It appears that the enumeration in 4.2 is a condensed repetition of the enumeration in 3.4. I think it does not add value, the methods should be compared in a single placed, and rather be described better.
* The quantitative analysis should be moved to the main paper, or at least be teasered there. For an explanation technique, it is really important to actually show the explanations to the readers (and reviewers).

**Requested Changes:**

Honestly, I think the required changes cannot be made in a revision within the given time limits. Nonetheless, I think the following things need to be changed:
1. Methodology - see my comments above - crucial
2. Experiments - see my comments above - crucial
3. Presentation - see my comments above - not crucial in the sense that only fixing these will not make me accept the paper, but crucial in the sense that without fixing these, the paper cannot be accepted.

---

### Review · Reviewer_EFS5 · 2026-03-22

**Summary Of Contributions:**

The paper proposes hyperparameter optimization using certain search strategies and random forests. They evaluate it on a small part of a single benchmark of surrogate models (2/14) which serve as replacement for machine learning models.

**Additional Comments:**

Just a minor revision for this paper will not improve it to standards of sufficient experimentation and sufficient interest.

**Audience:**

No

**Audience Explanation:**

They have an evaluation on a small set of the YAHPO benchmark https://arxiv.org/pdf/2109.03670. See Table 2 in there. They use 2 surrogates from the 14 present there, then train their own surrogate to approximate the surrogate, instead of using the given surrogate. But even if they would run it on all 14 surrogates, it would be still a toy test for small models of a straightforward, simple HPO algorithm.

One would be rather interested in evaluations of expensive models with a validated argument in the case if one would be able to skip certain hyperparameters.

This is a toy-type scenario of very little impact on applications of ML. The paper also does not contribute theory.

**Claims And Evidence:**

No

**Claims Explanation:**

- The setup is too brief which might be owed to the fact that they evaluate on surrogate models. Example: $y^{ref}$ is a performance. It is not said further what this quantity amounts to, whether it is a validation or training data statistic of some sort. Just a "performance".

  - Then they train a surrogate on a surrogate, additionally with almost no details. Section B in the appendix just says: hyperparameters are normalized to [0,1]. The original ranges are not given.

  - What is claimed to be counterfactuals are simple evaluations of a surrogate on a set of hyperparameters. To find hyperparameters, one need to evaluate either the real model or a surrogate anyway. Standard evaluation is not counterfactual generation.
  - This is misleading usage of the term or overselling.

  - the search methods are very simple: Random Restart "performs uniform sampling around the reference configuration" . "Random Search. It is a basic baseline method that performs completely random sampling"
    - How is completely random different from uniform sampling ?

  - The optimization objective (eq(1)) is unmotivated: if the performance would be a test loss oracle, then distance to start parameters would be of little interest. Why one would trade performance for closeness to a starting point ? Same holds for the loss components of proximity, diversity and sparsity ? As for sparsity, why one would care at all how much hyperparameter one needed to touch ? One wants best generalization. That is for most cases not a function of number of hyperparameters changed relative to an arbitrary starting point !


  - It also makes no sense to build a surrogate for another surrogate.

  - Also it is unclear what set the 90-percentile is taken over? Eq (4) simply has a big X. That big X summarizes the unclear theoretical notation of this paper well, unfortunately. Is this a training proxy  ? A validation proxy ?

  - Where does the variance come from that causes a set to emerge, over which then percentiles are measured ? Does it come from different training data ? If so, the extreme variances in table 1 in validity indicate some problem with the setting: either too little data, and bad proxies, or one uses very simple ML methods.

**Requested Changes:**

What is claimed to be counterfactuals seem to be simple evaluations of a surrogate on a set of hyperparameters. Thus keeping it in the title and paper as such is misleading.

The paper needs a complete overhaul.

Expand this paper to a scope that matters.
Example: pretraining of large LLMs or finetuning of those - a setup where skipping a step would save money. If one can show with theory that one can skip certain range under certain conditions, this would have some impact.

---

### Review · Reviewer_yVJT · 2026-04-13

**Summary Of Contributions:**

The submission proposes applying counterfactual explanations—typically used for model predictions in explainable AI—to the domain of Hyperparameter Optimization (HPO). The authors frame this as an optimization problem to identify "nearest/similar" hyperparameter settings that yield strong model performance. To circumvent the computationally expensive need for training multiple "counterfactual" configurations, the proposed CF-HPO framework utilizes proxy/surrogate models.

**Strengths:** The paper explores an interesting conceptual bridge between XAI and HPO. The literature review is solid, and the experimental setup using surrogate benchmarks is reasonable.

**Weaknesses:** The manuscript requires rigorous proofreading. Currently, the writing reads more like a course project report than a polished TMLR submission. There are numerous missing references, inconsistent formatting choices, and underdeveloped paragraphs. More importantly, the core conceptual motivation—specifically the need for a "distance function" over hyperparameters—lacks sufficient justification.

**Audience:**

Yes

**Audience Explanation:**

Yes. Members of the community working at the intersection of interpretability and optimization may find the idea interesting. However, the ultimate practical value to the broader audience may be limited unless the authors better justify _why_ counterfactual proximity matters in the context of hyperparameter tuning.

**Claims And Evidence:**

Yes

**Claims Explanation:**

Yes. The experimental setup is reasonable, and relying on surrogate benchmarks like YAHPO is a sound approach for evaluating the proposed framework without incurring exorbitant computational costs.

**Requested Changes:**

The following adjustments are required to strengthen the manuscript and address conceptual and formatting issues:

**Major Conceptual & Clarification Issues:**

- Relevance of Distance in HPO (Section 3.1): Why does distance from the reference point even matter? In the process of training models, the primary concern is ultimate performance (e.g., accuracy). Unless a hyperparameter choice has unintended side-effects (e.g., increasing the number of layers directly incurs more training time), distance from a reference configuration is not inherently meaningful. For example, a smaller/larger learning rate directly affects training dynamics based on its absolute magnitude, not relative to a reference value. Please justify this design choice.

- Counterfactual Criteria (Section 3.3): Why are proximity, diversity, and sparsity important in HPO? While these are standard, important criteria for counterfactual explanations in classification/prediction, it is entirely unclear why they are necessary objectives for hyperparameter tuning.

- Search Strategies (Section 3.3/3.4): What specific algorithm is being used to find the values for continuous parameters ("optimized using gradient updates")? - Where and by whom is this happening in the search for CFEs? Additionally, clarify the significance of the chosen methods (UCB, random restart, hill climbing) and how they work in this specific context.

- Categorization of Architecture: In Section 1, acknowledge that some practitioners argue model architecture is a hyperparameter itself.

- Distance Metrics (Section 2.2): The claim that mixed continuous/categorical variables "complicate distance calculations" is weak; there are existing distance metrics specifically designed to handle mixed data types.

**Writing, Formatting & Citations:**

- Abstract: Fix the typo "recomented" (should be "recommended"). Also, check the use of "generability"—consider replacing it with "generalizability".

- Introduction: The opening sentence ("Machine learning has been and continues to be the subject of thousands of studies using deep learning models.") is awkwardly phrased and unclear.

- Introduction: When discussing "Changes in parameters...", explicitly use the word "hyperparameters", as "parameters" typically refers to network weights.

- Introduction: Define the "HPO" acronym in the main text of the introduction, not just in the abstract.

- Introduction: The transition to "In these days of explainable artificial intelligence..." feels like a sudden and jarring change of story. Smooth this transition.

- Section 1: Clarify what is meant by altering a prediction "via optimization" when referring to Wachter et al.

- Section 1: Define what is meant by a "modular framework" when it is first introduced.

- Section 1: Avoid ending paragraphs with a colon (":"), especially if the text that follows consists of more paragraphs rather than a direct list.

- Section 2.1: There is a broken citation formatting near the end of the section ("...within a single framework.").

- Section 2.1: Regarding the natural-language description of configurations to avoid: Is this truly necessary? Why not just display the before/after numerical hyperparameter values?

- Section 2.2: Saying the approach was introduced to "the field of machine learning" is too broad; it is better to specify "explainable AI".

- Section 2.2: Remove the unnecessary line break after the Mothilal et al. (2020) citation.

- Section 2.2: The phrase "Integrating empirical explanations..." is strange and needs rephrasing.

- Missing Citations: Provide citations for the "YAHPO benchmark suite", "LCBench", and "IAML-Ranger" when introduced to assist unfamiliar readers.

- Page 5: Fix the missing space before the new sentence: "efficiency.Estimates".

---

### Decision · Action_Editor_P9FE · 2026-05-29

**Recommendation:** Reject

**Audience:**

No

**Audience Explanation:**

While reviewers agree the paper proposes a novel idea of using counterfactual explanations for HPO, their methodology requires a significant improvement to be useful for the TMLR's audience.

**Claims And Evidence:**

No

**Claims Explanation:**

All reviewers agree that this paper requires a major revision to provide justification to the motivation, more solid experimental results and a better writing.